# Relationships between body fat distribution and metabolic syndrome traits and outcomes: A mendelian randomization study

Brian Huang [1], John DePaolo [2], Renae L. Judy [2], Gabrielle Shakt [2], Walter R. Witschey [3], Michael G. Levin [4,5‡], Victoria M. Gershuni [2,6‡*]

1 Department of Medicine, Massachusetts General Hospital, Boston, MA, United States of America, 2 Department of Surgery, Hospital of the University of Pennsylvania, University of Pennsylvania, Philadelphia, PA, United States of America, 3 Department of Radiology, Perelman School of Medicine, University of Pennsylvania, Philadelphia, Pennsylvania, United States of America, 4 Division of Cardiovascular Medicine, Perelman School of Medicine, University of Pennsylvania, Philadelphia, PA, United States of America, 5 Corporal Michael J. Crescenz VA Medical Center, Philadelphia, PA, United States of America, 6 Perelman School of Medicine, University of Pennsylvania, Philadelphia, PA, United States of America

‡ MGL and VMG are jointly supervised this work.
* victoria.gershuni@uphs.upenn.edu

## Abstract

### Background

Obesity is a complex, multifactorial disease associated with substantial morbidity and mortality worldwide. Although it is frequently assessed using BMI, many epidemiological studies have shown links between body fat distribution and obesity-related outcomes. This study examined the relationships between body fat distribution and metabolic syndrome traits using Mendelian Randomization (MR).

### Methods/findings

Genetic variants associated with visceral adipose tissue (VAT), abdominal subcutaneous adipose tissue (ASAT), and gluteofemoral adipose tissue (GFAT), as well as their relative ratios, were identified from a genome wide association study (GWAS) performed with the United Kingdom BioBank. GWAS summary statistics for traits and outcomes related to metabolic syndrome were obtained from the IEU Open GWAS Project. Two-sample MR and BMI-controlled multivariable MR (MVMR) were performed to examine relationships between each body fat measure and ratio with the outcomes. Increases in absolute GFAT were associated with a protective cardiometabolic profile, including lower low density lipoprotein cholesterol (β: -0.19, [95% CI: -0.28, -0.10], p < 0.001), higher high density lipoprotein cholesterol (β: 0.23, [95% CI: 0.03, 0.43], p = 0.025), lower triglycerides (β: -0.28, [95% CI: -0.45, -0.10], p = 0.0021), and decreased systolic (β: -1.65, [95% CI: -2.69, -0.61], p = 0.0019) and diastolic blood pressures (β: -0.95, [95% CI: -1.65, -0.25], p = 0.0075). These relationships were largely maintained in BMI-controlled MVMR analyses. Decreases in

downloads.html and the IEU OpenGWAS Project (https://gwas.mrcieu.ac.uk/). OpenGWAS IDs are displayed in Supplementary Table 1.

**Funding:** J.D. is supported by the American Heart Association (23POST1011251). W.W. is supported by the NIH R01 (HL137984) and NIH P41 (EB029460) grants. M.G.L. is supported by the Institute for Translational Medicine and Therapeutics of the Perelman School of Medicine at the University of Pennsylvania, the NIH/NHLBI National Research Service Award postdoctoral fellowship (T32HL007843), and the Measey Foundation. V.M.G. is supported by the McCabe Foundation and the Institute for Translational Medicine and Therapeutics of the Perelman School of Medicine at the University of Pennsylvania, and the National Center for Advancing Translational Sciences of the National Institutes of Health under award number KL2TR001879. The content is solely the responsibility of the authors and does not necessarily represent the official views of the National Institutes of Health. The funders had no role in study design, data collection and analysis, decision to publish, or preparation of the manuscript.

**Competing interests:** The authors have declared that no competing interests exist.

relative GFAT were linked with a worse cardiometabolic profile, with higher levels of detrimental lipids and increases in systolic and diastolic blood pressures.

## Conclusion

A MR analysis of ASAT, GFAT, and VAT depots and their relative ratios with metabolic syndrome related traits and outcomes revealed that increased absolute and relative GFAT were associated with a favorable cardiometabolic profile independently of BMI. These associations highlight the importance of body fat distribution in obesity and more precise means to categorize obesity beyond BMI.

## Introduction

Obesity is a complex, multifactorial disease that has greatly increased in prevalence worldwide. Recent estimates predict that 2.5 billion people globally are either obese or overweight, comprising 39% of the world population [1]. Excess adiposity is associated with a number of adverse health outcomes, including diabetes [2], cardiovascular disease [3], cancer [4], chronic kidney disease [5], and musculoskeletal conditions [6], among others. One well-known adverse outcome is metabolic syndrome, a constellation of cardiovascular risk factors comprised of hypertension, hyperlipidemia, insulin resistance, and increased weight circumference [7]

Body mass index (BMI, kg/m$^2$) is the primary metric used to define and identify overweight and obese individuals. While BMI measurements are easily measured and collected, they are unable to capture the variation in body composition between individuals, such as differences between lean and fat mass and the varying distribution of fat depots within the body (i.e. visceral, subcutaneous, gluteofemoral) [8]. This is of particular importance given the recognition of a "metabolically healthy obesity" phenotype, thought to be driven in part by the more favorable metabolic activity of gluteofemoral fat compared to visceral fat [9]. Many studies have sought more precise measures of body fat to better identify and stratify those at higher risk of obesity and overweight-related complications.

Central obesity has been widely studied as a particularly significant risk factor for poor cardiometabolic outcomes such as insulin resistance [10], diabetes [11], hypertension [12], and coronary artery disease [13] in different populations. Individuals with high central obesity but normal BMI were also found to have higher mortality than BMI-defined obesity, particularly when the BMI-based obese individuals had less central fat [14]. Conversely, fat in the gluteal and femoral regions has been observed to have a potentially protective effect for obesity-related outcomes. Epidemiological studies on gluteofemoral adipose tissue (GFAT) have shown associations with decreased insulin resistance [15], lower rate of myocardial infarction [16], and a heathier lipid profile [17], among others.

Although obesity is influenced by numerous social, environmental, and behavioral factors, the heritability of obesity has been estimated to be between 40 and 70% in epidemiological studies, indicating a substantial genetic component to the disease [18, 19] With the advancements in molecular genotyping over the past few decades, population-scale genome-wide association studies (GWAS) of BMI and other anthropometric measures have offered substantial insight into the genetic basis of obesity [20]. Recently, GWAS have begun to examine the genetics of adipose distribution. A recent study performed a large-scale GWAS of MRI-measured GFAT, abdominal subcutaneous adipose tissue (ASAT), and visceral adipose tissue

(VAT), recapitulating multiple previous obesity-related loci while also identifying new genetic loci for the different fat depots [21].

Mendelian randomization (MR) is a technique utilizing SNPs identified on GWAS as instrumental variables to better establish causal relationships between modifiable risk factors and outcomes compared to conventional epidemiological studies. Since genetic variants are randomly spread throughout the population through the natural independent assortment during meiosis, they are unlikely to be generally associated with the numerous common behavioral and social confounders that can limit more traditional epidemiological analyses [22]. In this study, we explored the relationship between body fat distribution and cardiometabolic profiles by using two-sample MR to examine the effects of ASAT, GFAT, and VAT, along with their relative ratios, on cardiometabolic biomarkers and outcomes, focusing specifically on those related to metabolic syndrome.

## Methods

### Fat depot GWAS

Genetic variants associated with GFAT, VAT, and ASAT were identified from a GWAS performed by Agrawal et al. [21] The GWAS was performed among 39,076 participants of the UK Biobank who underwent abdominal MRI scans [23]. Direct measurements of VAT and ASAT were available for approximately 9000 patients, and GFAT was estimated by subtracting VAT and ASAT from the total measured fat volume between the thighs and T9 vertebrae, which was recorded for about 7750 patients. Agrawal et al. trained a convolutional neural network machine learning model on this dataset with an 80/20 training and holdout split. Five-fold cross validation was performed to assess model performance on the training set, and the model was evaluated using $r^2$ on the holdout test set, achieving high $r^2$ values of 0.991, 0.991, and 0.978 for VAT, ASAT, and GFAT respectively. The deep learning model was then applied to estimate the VAT, ASAT, and GFAT volumes in the remaining patients with MRI scans. Full details regarding the specifics of the convolutional neural network architecture and parameters can be found within the original study [24]. ASAT/GFAT, VAT/ASAT, and VAT/GFAT values were computed as a direct ratio of predicted volumes.

Each trait was inverse-normal transformed prior to performing GWAS. Complete parameters used to filter genotypes and perform quality control can be found within the original manuscript. Lead SNPs were prioritized using LD clumping (p1: 5e-8, p2: 5e-6, $r^2$: 0.1, distance: 1000 kb) with a reference panel constructed from a random sample of 3000 individuals from the overall study set. The lead SNPs identified for each of the six traits (ASAT, GFAT, VAT, ASAT/GFAT, VAT/ASAT, and VAT/GFAT) were extracted from the overall GWAS summary statistics. Overall $r^2$ was calculated for each set of lead SNPs for each trait.

### Cardiometabolic outcomes GWAS

Metabolic syndrome is commonly defined as three or more of the following traits: abdominal obesity, increased blood pressure, increased blood glucose, increased triglycerides, and decreased HDL-cholesterol (HDL-c). Outcomes were selected based on these traits and included LDL cholesterol (LDL-c) [25], Apolipoprotein-B (Apo-B) [26], HDL-c [25], Apolipoprotein A-1 (Apo-A1) [26], total cholesterol [25]), total triglycerides [25], systolic blood pressure (SBP) [27], diastolic blood pressure (DBP) [27], fasting glucose [28], HbA1c [29], and type 2 diabetes (T2DM) [30]. GWAS summary statistics for each outcome were obtained from IEU Open GWAS Project (https://gwas.mrcieu.ac.uk/), selecting studies with larger study populations when multiple sets of summary statistics were available for an outcome. A full summary of the individual studies used for outcome GWAS can be found in S1 Table.

## Mendelian randomization

All MR analysis was carried out using the *TwoSampleMR* package in R (https://mrcieu.github.io/TwoSampleMR/index.html) following the STROBE-MR guidelines [31, 32]. The primary MR analysis was performed using inverse variance weighting (IVW) with multiplicative random effects. MR using Weighted median and MR-Egger were performed as sensitivity analyses, as these methods make different assumptions about the presence of pleiotropy which could lead to biased estimates of causal effects [33]. The MR-Egger intercept test was performed to evaluate for the presence of horizontal pleiotropy. Leave-one-out and single-SNP analyses were performed and visually inspected to identify the presence of outlier or influential variants.

## Multivariable mendelian randomization

Multivariable mendelian randomization (MVMR) is an extension of MR to more explicitly account for potential pleiotropic pathways, and estimate the direct effect of an exposure on an outcome, conditional on other exposures. Since fat distribution may be influenced by BMI, MVMR was performed to evaluate the effect of the individual fat depots while controlling for BMI. Significant SNPs for each body depot mass were harmonized with the largest BMI GWAS from the OpenGWAS project (https://gwas.mrcieu.ac.uk/datasets/ieu-b-40/) [34]. The effect of each adipose trait on each cardiometabolic outcome was then evaluated using the *TwoSampleMR* package in R.

## Statistical analysis

All analyses were performed using R (version 4.2.0; R Foundation for Statistical Computing, Vienna, Austria). P-values less than 0.05 were considered statistically significant.

# Results

## Univariate mendelian randomization

**Absolute body fat depot measurements.** The association between increased adiposity within each depot (ASAT, GFAT, and VAT) and each metabolic syndrome outcome was first estimated using univariate MR. Fig 1 contains the complete results of the univariate MR analyses examining the relationship between genetically instrumented ASAT, GFAT, and VAT and each continuous cardiometabolic outcome while S1 Fig contains the results of the MR analyses examining T2DM. Increased levels of GFAT were significantly associated with lower LDL-c (β: -0.19, [95% CI: -0.28, -0.10], p < 0.001) and Apo-B levels (β: -0.18, [95% CI: -0.24, -0.12], p < 0.001), along with higher HDL-c (β: 0.23, [95% CI: 0.03, 0.43], p = 0.025) and Apo-A1 levels (β: 0.20, [95% CI: 0.06, 0.34], p = 0.0048). Increased GFAT was also associated with significantly lower levels of total cholesterol (β: -0.16, [95% CI: -0.26, -0.06], p = 0.0015) and triglycerides (β: -0.28, [95% CI: -0.45, -0.10], p = 0.0021). Neither genetically instrumented ASAT or VAT were significantly associated with any lipid outcome in the univariate analysis.

For diabetes outcomes, GFAT was associated with lower fasting glucose (β: -0.04, [95% CI: -0.08, -0.01], p = 0.012), but there was not a significant trend for A1c levels and T2DM risk. ASAT was significantly associated with increased T2DM risk (OR: 3.15, [95% CI: 1.74, 5.72], p <0.001), while VAT was associated with a lower fasting glucose level (β: -0.08, [95% CI: -0.13, -0.02], p = 0.0068). GFAT was also associated with decreased SBP (β: -1.65, [95% CI: -2.69, -0.61], p = 0.0019) and DBP (β: -0.95, [95% CI: -1.65, -0.25], p = 0.0075), while ASAT and VAT were not significantly associated with either blood pressure measurement. The $r^2$ values for each set of lead SNPs for ASAT, GFAT, and VAT were 0.0050, 0.0172, and 0.0041

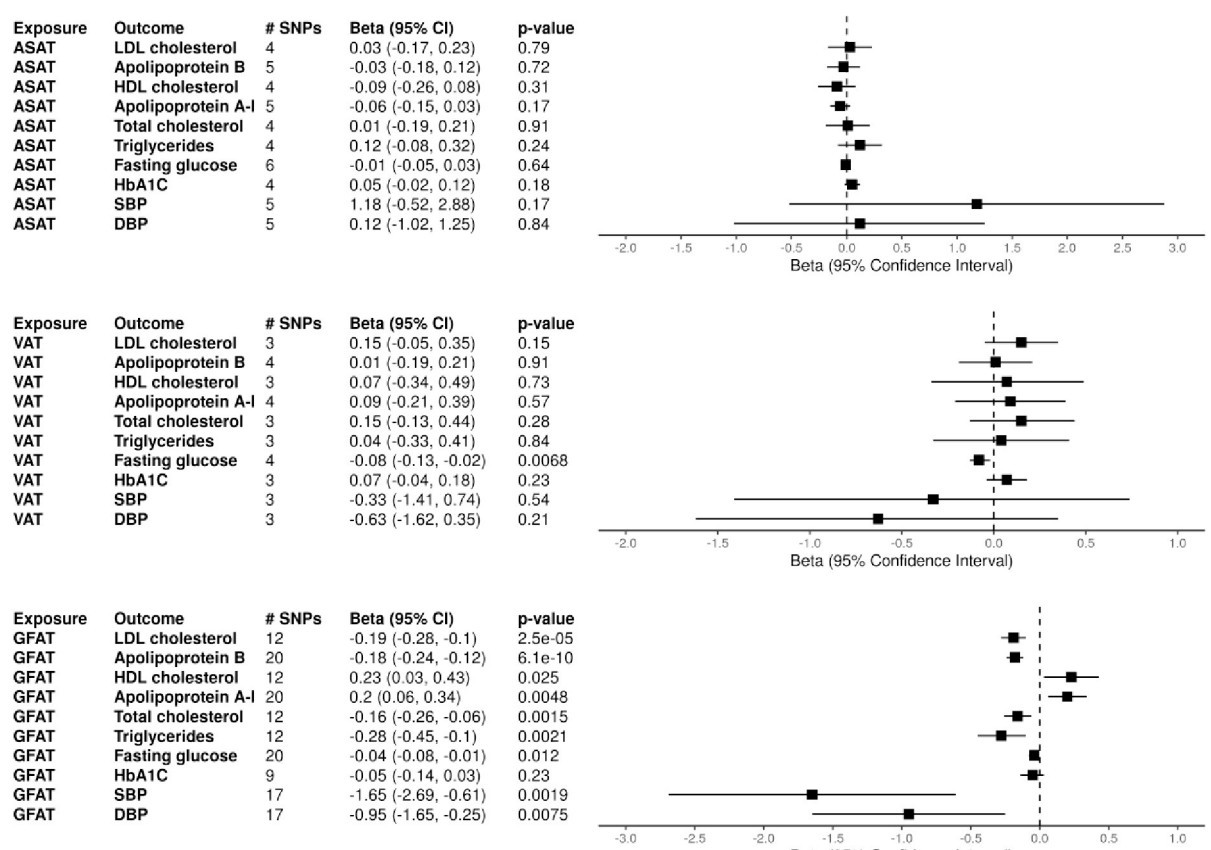

**Fig 1. Univariate mendelian randomization with absolute fat depots.** Two sample mendelian randomization (MR) results with genetically instrumented fat depots as exposures and cardiometabolic markers outcomes are shown. MR was performed with inverse-variance weighted estimates. Number of SNPs extracted from each exposure GWAS as instruments are shown as # SNPs. Abbreviations: ASAT (abdominal subcutaneous adipose tissue), GFAT (gluteofemoral adipose tissue), VAT (visceral adipose tissue), SBP (systolic blood pressure), DBP (diastolic blood pressure).

respectively. The harmonized datasets used to perform the analyses with absolute fat depot measurements are displayed in S2-S4 Tables of S1 File.

**Body fat depot ratios.** To better understand how the relative ratios of the three fat depots affect cardiometabolic outcomes, we repeated the univariate analysis with the genetically instrumented ASAT/GFAT, VAT/ASAT, and VAT/GFAT ratios as exposures. For the lipid outcomes, a higher ASAT/GFAT ratio was associated with significantly increased LDL-c (β: 0.11, [95% CI: 0.03, 0.19], p = 0.0076), Apo-B (β: 0.12, [95% CI: 0.04, 0.21], p = 0.004), and total triglycerides (β: 0.32, [95% CI: 0.24, 0.41], p < 0.001), along with decreased HDL-c (β: -0.31, [95% CI: -0.39, -0.23], p < 0.001) and Apo-A1 levels (β: -0.22, [95% CI: -0.31, -0.13], p < 0.001). A higher VAT/GFAT ratio was associated with the same outcomes, with increased LDL-c (β: 0.11, [95% CI: 0.02, 0.19], p = 0.19), Apo-B (β: 0.16, [95% CI: 0.11, 0.21], p < 0.001), and total triglycerides (β: 0.30, [95% CI: 0.15, 0.45], p < 0.001), and lower HDL-c (β: -0.23, [95% CI: -0.42, -0.04], p = 0.016) and Apo-A1 (β: -0.15, [95% CI: -0.28, -0.03], p = 0.017). An increased VAT/ASAT ratio was associated with an increased Apo-B (β: 0.09, [95% CI: 0.05, 0.13], p < 0.001) and total triglyceride level (β: 0.15, [95% CI: 0.02, 0.28], p = 0.022).

Higher ASAT/GFAT ratio was associated with both increased HbA1C (β: 0.08, [95% CI: 0.04, 0.12], p < 0.001) and T2DM (OR: 1.59, [95% CI: 1.10, 2.30], p = 0.013), while VAT/GFAT was not associated with any diabetes outcome. Increased VAT/ASAT was associated with

increased risk of T2DM (OR: 1.39, [95% CI: 1.05, 1.85], p = 0.023), but not glucose or HbA1C. ASAT/GFAT ratio was also associated with an increased SBP (β: 1.37, [95% CI: 0.16, 2.58], p = 0.026), while VAT/GFAT ratio was associated with an increased DBP (β: 1.32, [95% CI: 0.26, 2.37], p = 0.014). Fig 2 displays the full set of results from the univariate body ratio MR analyses on continuous outcomes and S1 Fig shows the results of the T2DM MR analyses. Harmonized datasets used for the body fat depot ratio analyses are shown in S5-S7 Tables of S1 File.

**Univariate sensitivity analyses.** MR with a weighted median estimate yielded largely consistent results for each of the univariate analyses. Leave-one-out analyses and single SNP analyses yielded similar estimates and directionality for each pair of exposures and outcomes. MR-Egger intercept testing did not detect any evidence of directional pleiotropy (p > 0.05) for any pair of exposure and outcomes apart from VAT/ASAT and ApoB. Forest plots for the weighted median, leave-one-out, and single SNP analyses are displayed in S2/S3, S4, and S5 Figs respectively.

**Multivariable Mendelian Randomization (MVMR).** Body fat distribution is heterogenous and individual fat depots are heavily dependent on overall body fat mass. As such, ratios between body fat depots may represent an incomplete measure of overall body fat distribution; thus, we performed MVMR to examine the effects of the individual fat depots controlled for BMI on cardiometabolic outcomes.

Genetically instrumented GFAT controlled for BMI had associations with lipid measures that were largely consistent with the univariate analyses, with higher levels of GFAT associated with lower levels of LDL-c, Apo-B, total cholesterol, and total triglycerides. GFAT controlled for BMI was also associated with increased Apo-A1 levels, but the estimate for HDL, while in the same direction as the univariate analyses, was not significant. BMI-controlled ASAT was associated with higher Apo-B levels, while BMI-controlled VAT was significantly associated with both increased HDL-c and total cholesterol.

For diabetes-related outcomes, BMI-controlled GFAT was significantly associated with decreased fasting glucose and HbA1C levels, along with decreased risk for T2DM. BMI-controlled ASAT was not associated with any diabetes outcomes. MVMR was not able to performed for BMI-controlled VAT and T2DM due an insufficient number of shared SNPs and the lack of a proxy SNP within BMI. Full results for the MVMR can be found within Figs 3 and S1.

## Discussion

Obesity is a nuanced and complicated disease, with numerous contributing social and genetic factors along with a large array of potential outcomes. Prior observational studies evaluating the effects of obesity on features of metabolic syndrome have often focused on BMI, a crude measure of obesity, and may have been further limited by residual confounding or reverse causality [35–37]. Here, we leveraged a recent large GWAS of the different metabolically relevant fat depots in both the trunk and the lower body to examine the relationship between the accumulation of fat within specific depots and cardiometabolic biomarkers related to the metabolic syndrome. This is the first MR study performed utilizing genetically instrumented fat depots to examine these outcomes. Overall, we find that regional adipose depots are associated with distinct metabolic profiles, which may have important implications for the diagnosis and treatment of obesity and related cardiometabolic diseases.

The results most markedly show that increased GFAT is generally associated with a more protective cardiometabolic profile. These findings were consistent across different features of the metabolic syndrome. For example, we found that increased GFAT was associated with a protective lipoprotein profile (lower levels of LDL-c/Apo-B, total cholesterol and triglycerides, and increased levels of HDL-c/Apo-A1), improved blood pressure, and improved fasting

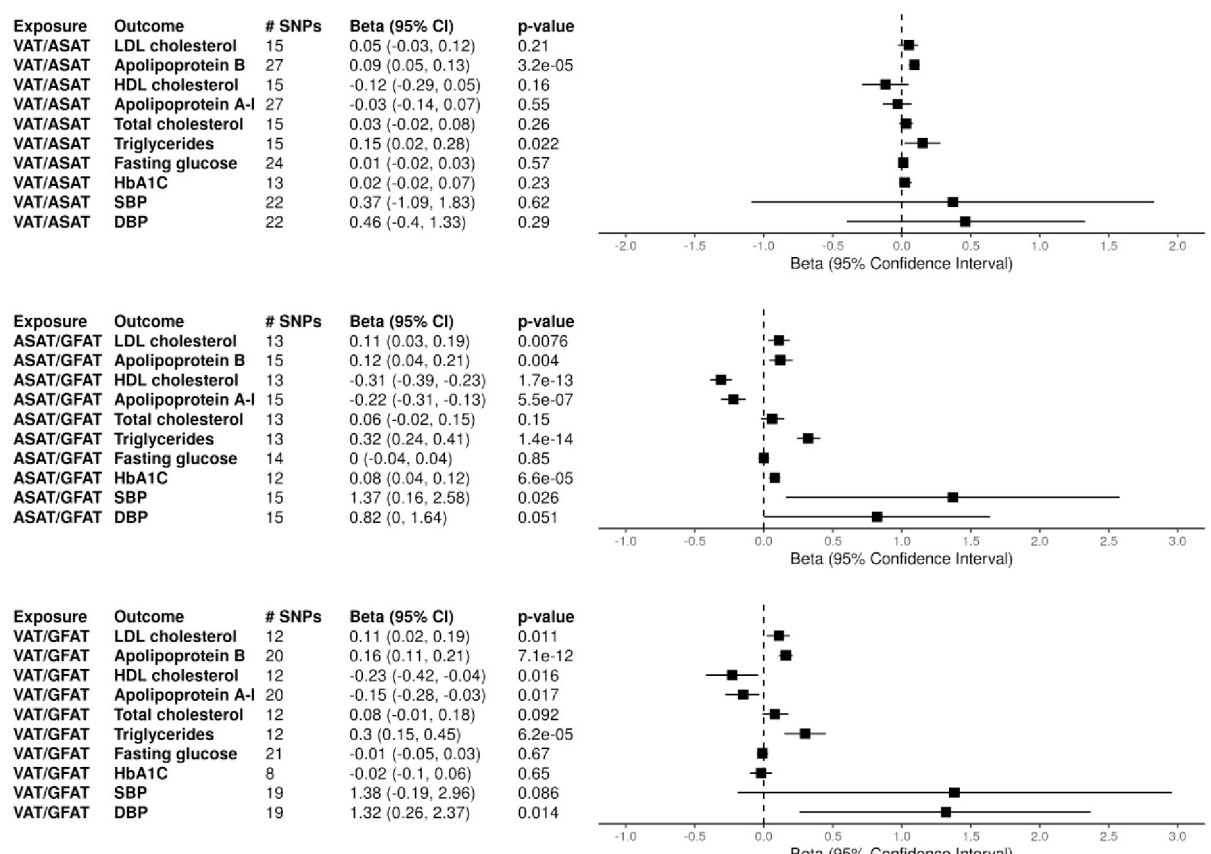

**Fig 2. Univariate mendelian randomization with relative fat depot ratios.** Two sample mendelian randomization (MR) results with genetically instrumented fat depot ratios as exposures and cardiometabolic markers as outcomes are shown. MR was performed with inverse-variance weighted estimates. Number of SNPs extracted from each exposure GWAS as instruments are shown as # SNPs. Abbreviations: ASAT (abdominal subcutaneous adipose tissue), GFAT (gluteofemoral adipose tissue), VAT (visceral adipose tissue), SBP (systolic blood pressure), DBP (diastolic blood pressure).

glucose. These results were largely consistent when examining body fat distribution ratios, where increased ASAT/GFAT and VAT/GFAT were associated with a higher risk cardiometabolic profile. Both ratios were significantly associated with less protective lipoprotein levels, with higher LDL-c/Apo-B and total triglycerides, along with lower HDL-c/Apo-A1 levels. ASAT/GFAT was additionally associated with higher SBP, HbA1c and diabetes risk, while VAT/GFAT was additionally associated with higher DBP. The associations between GFAT and outcomes remained consistent and significant when controlling for BMI with MVMR with the exceptions of HDL-c and DBP. The MVMR analyses additionally demonstrated a stronger decreased risk association between GFAT and diabetes, with lower fasting glucose, HbA1c, and T2DM risk. In combination, these results suggest that subjects who have both absolute and relative increases in fat distributed to the GFAT depot may have beneficial cardiometabolic effects.

The protective effect of absolute and relative GFAT observed within this study is consistent with prior observational studies. One study of 755 individuals in a Caucasian European population found that an increased waist-hip ratio (WHR), a proxy for decreased GFAT and increased ASAT and VAT, was associated with higher triglycerides and cholesterol, along with lower FFA and HDL-c. WHR is most closely related to the ASAT/GFAT and VAT/GFAT ratios examined in this study, which similarly demonstrated a relationship between relatively

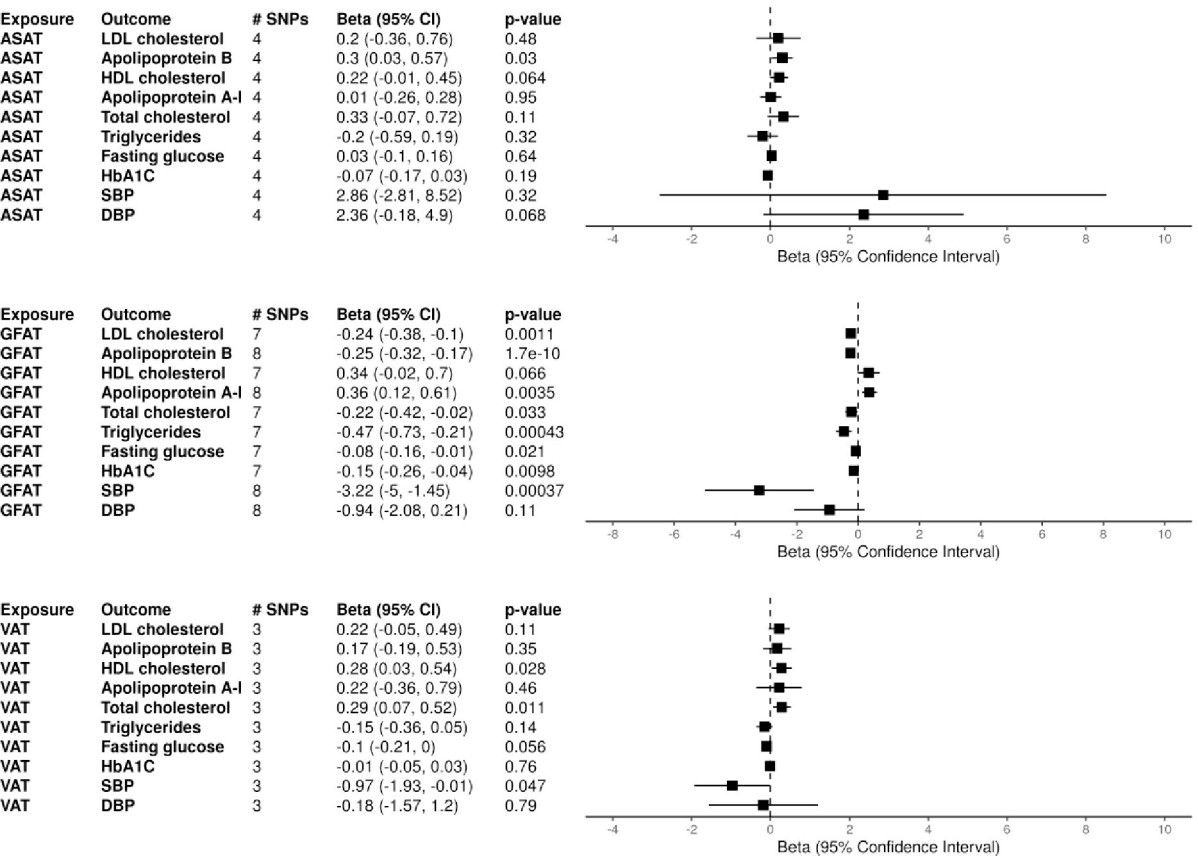

**Fig 3. Multivariable mendelian randomization results controlled for BMI.** Multivariable mendelian randomization controlling for BMI was performed with absolute fat depots as exposures and cardiometabolic markers as outcomes and shown. Genetic instruments for each fat depot were harmonized with the BMI GWAS from the GIANT consortium [34]. The number of genetic instruments for each depot is displayed as # SNPs. Abbreviations: ASAT (abdominal subcutaneous adipose tissue), GFAT (gluteofemoral adipose tissue), VAT (visceral adipose tissue), SBP (systolic blood pressure), DBP (diastolic blood pressure).

decreased GFAT and increased triglycerides and cholesterol, along with lower HDL-c. Waist circumference (WC) and hip circumferences also showed significant opposing effects on tri-glycerides, cholesterol, and HDL-c, with increased hip circumference appearing cardio-meta-bolically protective [15]. Since GFAT is a major component of hip circumference, these epidemiological protective associations are supported by the results from this study.

Another study examined CT-measured subcutaneous thigh fat in 2106 participants aged 70–79 and found that higher amounts were associated with lower triglycerides and higher HDL-c after being adjusted for abdominal subcutaneous, visceral, and intramuscular thigh fat [38]. While thigh fat is only a component of GFAT, these results are similar to those in this study and support a potential independent impact of this fat depot on triglycerides and HDL-c. A smaller study of 316 individuals using MRI-measured fat depots found similarly that GFAT was associated with decreased insulin and triglycerides, but did find that this association disappeared when controlled for VAT [15]. In contrast, our associations with fasting glucose and triglycerides remained when controlled for BMI, although this has many additional con-tributing factors beyond VAT.

A few previous genetic studies have also been conducted examining the relationship between obesity and a range of cardiometabolic outcomes, although have largely focused on proxy anthropometric phenotypes, such as WHR and WC, rather than directly measured fat

depots. For example, several prior studies have used polygenic risk scores to examine the associations between anthropometric measures of obesity and some features of the metabolic syndrome. These analyses have found links between increased BMI and WHR with increased atherogenic lipid profiles and blood pressure, as well as an increased risk for T2DM and coronary artery disease [39–41]. More recently, MR has been used to estimate the effects of GFAT, BMI and WC on glycemic traits, finding that increased WC and WC-adjusted BMI were associated with increased fasting insulin levels, although a significant effect of GFAT on fasting insulin was not found [42]. While these studies have found associations between anthropometric measures of body fat distribution and metabolic syndrome features, our results suggest that regional adipose depots may have direct and distinct effects on these outcomes. Previous work has demonstrated deleterious metabolic associations with WHR and WC, both of which have a component of abdominal adiposity; this study instead highlights protective associations of GFAT on similar outcomes of lipids, blood pressure, and diabetes.

The observed protective effect of GFAT has been attributed to the different metabolic activity of the adipocytes within different fat depots. These mechanisms primarily encompass assorted site-specific responses to hormone signaling and secretion of adipose-related tissue proteins (adipokines) [43]. Mechanistically, adipocytes vary in behavior and function with site-specific expression of early developmental genes, suggestive of differential embryonic origins, directing adipose tissue development [44]. This is particularly relevant to the discovery of a "metabolically healthy" phenotype where the distribution of body fat, rather than overall adiposity, seems to impact cardiometabolic consequences. This is presumably caused by the ability of lower body adipocytes to undergo hyperplasia, leading to increased lipid storage capacity and decreased non-esterified fatty acid release. In comparison, abdominal adipocytes undergo hypertrophy, resulting in more limited storage capacity and increased lipolysis [9].

As such, abdominal fat, inclusive of both ASAT and VAT, has a high lipid turnover and is the primary site for diet-derived fat. Conversely, GFAT undergoes less cellular triglyceride turnover, less triglyceride uptake, and is less sensitive to adrenergic lipolysis stimulation, causing greater FA retention within GFAT and decreased ectopic fat deposition (i.e. in the liver). These trends in lipolysis were supported by multiple in vitro studies. An early study examining fat samples taken from a small number of participants found that abdominal adipocytes greatly increased rate of lipolysis compared to gluteal fat cells when exposed to noradrenaline [45]. A more recent experiment demonstrated that gluteal adipose tissue had a lower rate of action for hormone-sensitive lipase, an important enzyme in lipolysis [42]. Another contributing protective mechanism for GFAT is the production of a higher proportion of palmitoleic acid, which has been proposed to be an insulin-sensitizing lipokine, compared to ASAT [46, 47]. Increased GFAT has also been associated with higher levels of adiponectin, which in turn is correlated with higher insulin sensitivity, better lipid profiles, and reduced inflammation [17, 48, 49]. This protective effect is highlighted in the context of familial lipodystrophies where genetic loss of GFAT is associated with metabolic derangement with greater cardiometabolic risk [50].

The associations between both absolute and relative GFAT levels and protective cardiometabolic biomarkers found in this study build upon and extend these previous observational and genomic-based studies. Compared to previous work, this study examined a wider range of outcomes encompassing more components of metabolic syndrome, including lipids, blood pressure, and diabetes. One strength of this analysis is that it employed genetic instruments for direct measures of ASAT, GFAT, and VAT and corresponding ratios instead of proxies such as WHR or WC, allowing for a more direct assessment of the effects from individual fat depots. This study also leverages the large quantity of available genetic data, examining these relationships in a much larger population compared to prior observational studies. Another major strength of this study is the ability to provide evidence towards a causal relationship through

the MR methodology independent of potential common confounders, such as environmental or behavior risk factors for obesity, strengthening the validity of the protective effects found through both epidemiologic and mechanistic studies. While the current paradigm for identifying overweight and obese individuals is still based on BMI, the results from this study, along with the existing literature, highlight that body fat distribution plays a large role in cardiometabolic risk and could be used to create more discriminative tools and groups when evaluating obesity.

## Limitations

This study should be interpreted within the context of its limitations. First, the pathophysiology behind the protective cardiometabolic associations with GFAT is not able to be fully elucidated. These observed relationships could be related to the distinct metabolic properties and hormone responsiveness of GFAT resulting in decreased fatty acid release; however, this study is not able to make a direct connection between the results and the hypothesized mechanisms. Furthermore, it is unclear whether some of the benefits are derived from the absence of fat deposited in more harmful adipose pockets and the hormonal and metabolic milieu associated with their deposition; it is possible that a strict increase in GFAT does not confer the same benefits. There is also no evidence that adipose tissue can be preferentially deposited through lifestyle changes or pharmacologic interventions, so GFAT may not represent a modifiable risk factor or intervention target. Further studies to assess for the effect of an isolated GFAT increase would be useful to establish a stronger causal relationship and evaluate for therapeutic potential.

Similarly, this study is not able to provide further mechanistic insight into the observed association between ASAT and T2DM, which could be from ASAT representing a general increase in adiposity or the specific metabolic features of ASAT itself. Additionally, these results did not demonstrate strong associations between the measures of central adiposity (VAT and ASAT) and the other markers of metabolic syndrome; however, due to wide confidence intervals surrounding these effect estimates, we cannot exclude potentially meaningful effects of these depots, particularly given the extensive literature documenting the negative effects of central adiposity. WC, a proxy for central adiposity, and directly measured VAT have both been associated with poor cardiometabolic outcomes, including hypertension, dyslipidemia, CAD, and T2DM [51]. A recent large CT study of abdominal traits in 13,422 patients found strong and highly significant associations between VAT and ASAT with diabetes mellitus, hypertension, chronic kidney disease, renal failure, and renal transplant, as well as a significant increase in triglycerides, glycated hemoglobin, blood urea nitrogen, creatinine, and BMI [52].

There are multiple reasons why we may not have observed these associations in this analysis. Firstly, the $r^2$ values for ASAT and VAT were low, indicating that there is large amount of variation within ASAT and VAT values that are not attributable to the lead SNPs. This is likely due to the high polygenicity of obesity as well as the substantial environmental influence on the development of fat depots. Diet, exercise, and multiple different social determinants of health including income, food availability, and transportation are all important components to obesity that are not addressed within this analysis [53]. Unaccounted genetic pathways linking central adiposity and adverse outcomes or these environmental contributions to the epidemiologically observed relationships could both explain the lack of their detection within this analysis. Another possibility is a shared upstream genetic pathway between abdominal adiposity and adverse outcomes rather than a direct effect causing their observed associations. Additionally, there may be significant fat volume variation that is genetically driven but not outside expected ranges for each depot and as such may not demonstrate the same relationships as a

true pathologic increase in central adiposity. Finally, the current MRI measures of VAT and ASAT depots may not capture the full extent of abdominal adiposity, such as ectopic fat depots within the muscles or liver, and therefore underestimate the negative effect on the outcomes. Future analyses using larger GWAS of these adipose depots to capture a larger proportion of the genetic variation, or more complete measurements of central adipose tissue may better capture the expected epidemiological relationships for central adiposity. Additional work examining the role that environmental factors may play in the relationships between each of these fat depots and cardiometabolic outcomes would also be beneficial and may offer further insight into potential avenues or tools to target modification in body fat distribution.

The mechanisms behind the relationships seen in analysis of the fat depot ratios are also unable to be identified with this analysis. The effects observed were mostly aligned with the results from each depot, suggesting they are primarily driven by the effects of the individual depots themselves. However, there is also potential for an additional interactive effect causing the ratio itself to have a differential impact, particularly given the more recently discovered adipokine signaling pathways influencing fat depot metabolism. Prior work regarding this interaction has been limited and would be another interesting avenue of further exploration [43, 45].

Next, the exposure and outcome data underwent different degrees of transformation and normalization before each GWAS, making a direct clinical interpretation of the MR estimate values difficult. As such, only directional conclusions can be drawn about the cardiometabolic outcomes. An additional limitation is that the GWAS of adipose depots used to derive genetic instruments consisted exclusively of individuals from ages 40–69, with 94% of individuals self-identifying as Caucasian [21]. While the sample size is large and representative of a relatively healthy UK population, the results may not be generally applicable to all patient groups, particularly given the complex relationships between age, race/ethnicity, and obesity [54, 55]. Further studies should be performed with more diverse cohorts to better assess the generalizability of these conclusions.

Finally, while MR studies are generally less susceptible to confounding than epidemiological work, it relies on three main validity assumptions: the genetic variants must be associated with the exposures, the genetic variants should not be associated with any confounding variables, and the genetic variants should not create any causational effect on the outcome apart from through the risk factor. Since strong genetic instruments were selected ($p < 5x10^{-8}$ for each significant SNP) from the GWAS for the fat depots and ratios, there is high confidence in an association between the variants and exposures. While the second and third assumptions are difficult to directly evaluate, the sensitivity analyses showed low amounts of variability in estimates and there was no evidence of pleiotropy in MR-Egger intercept tests for nearly all of the findings. Confounding by an additional outcome correlated with the genetic instruments cannot be excluded, but is also less likely given the results of the sensitivity analyses.

In summary, this study performed the first MR analysis between the ASAT, GFAT, and VAT fat depots, along with their relative ratios, and an array of cardiometabolic traits related to the metabolic syndrome. The results demonstrated that both absolute and relative increases in GFAT were associated with a favorable cardiometabolic profile, and these effects were largely independent of BMI. This work builds on previous observational findings, and by using the MR framework provides stronger evidence to support a causal relationship. The associations between individual fat depots and outcomes highlights the significance of regional adiposity and body fat distribution when evaluating cardiometabolic risk in individuals and supports the notion that more precise methods to categorize obesity beyond BMI are needed.

## Supporting information

**S1 Table. GWAS study table for outcome traits.** Details regarding the GWAS studies referenced for each cardiometabolic outcome trait are displayed. Abbreviations: GLGC (Global Lipids Genetics Consortium), UKB (United Kingdom BioBank), MAGIC (Meta-Analyses of Glucose and Insulin-related traits Consortium), DIAGRAM (Diabetes Genetics Replication And Meta-analysis), GERA (Genetic Epidemiology Research on Adult Health and Aging), ICBP (International Consortium for Blood Pressure).
(DOCX)

**S1 File. Harmonized data for univariate MR.** Dataframes containing harmonized data used for univariate mendelian randomization with cardiometabolic traits as outcomes. Variant data was extracted from each exposure GWAS and harmonized with each outcome GWAS.
(XLSX)

**S1 Fig. Mendelian randomization results for type 2 diabetes.** Results for univariate mendelian randomization (MR) with absolute fat depot values and their relative ratios as exposures and type 2 diabetes (T2DM) as the outcome are shown in a). Results for multivariable mendelian randomization (MVMR) for the absolute fat depots controlling for BMI are shown in b). MR and MVMR estimates were calculated with inverse variance weighting and logarithmic scaling was applied to the x-axis to display confidence intervals. # SNPs denotes the number of SNPs used as instruments from each exposure GWAS. Abbreviations: ASAT (abdominal subcutaneous adipose tissue), GFAT (gluteofemoral adipose tissue), VAT (visceral adipose tissue).
(DOCX)

**S2 Fig. Mendelian randomization results for absolute fat depots with weighted median estimates.** Results for univariate mendelian randomization (MR) with absolute fat depots as exposures and cardiometabolic markers as outcomes using weighted median estimates. Logarithmic scaling was applied to the x-axis for the Type 2 diabetes outcome MR. Abbreviations: ASAT (abdominal subcutaneous adipose tissue), GFAT (gluteofemoral adipose tissue), VAT (visceral adipose tissue). SBP (systolic blood pressure), DBP (diastolic blood pressure.
(DOCX)

**S3 Fig. Mendelian randomization results for fat depot ratios with weighted median estimates.** Results for univariate mendelian randomization (MR) with relative fat ratios as exposures and cardiometabolic markers as outcomes using weighted median estimates. Logarithmic scaling was applied to the x-axis for the Type 2 diabetes outcome MR. Abbreviations: ASAT (abdominal subcutaneous adipose tissue), GFAT (gluteofemoral adipose tissue), VAT (visceral adipose tissue). SBP (systolic blood pressure), DBP (diastolic blood pressure).
(DOCX)

**S4 Fig. Leave-one-out analyses.** Inverse variance weighted mendelian randomization estimates excluding the indicated SNP were calculated for each SNP and pair of exposures and outcomes. VAT on T2DM leave-one-out analyses was unable to be performed due to insufficient number of genetic instruments for VAT. Estimates obtained with all SNPs included are also displayed at the bottom of each graph. ASAT, VAT, GFAT, VAT/ASAT, ASAT/GFAT, and VAT/GFAT are shown in parts a), b), c), d), e), and f) respectively. Abbreviations: ASAT (abdominal subcutaneous adipose tissue), GFAT (gluteofemoral adipose tissue), VAT (visceral adipose tissue). SBP (systolic blood pressure), DBP (diastolic blood pressure.
(DOCX)

**S5 Fig. Single SNP sensitivity analysis.** Inverse variance weighted mendelian randomization estimates were computed with single SNPs for each SNP and pair of exposures and outcomes. Estimates obtained with all SNPs included are also displayed at the bottom of each graph. ASAT, VAT, GFAT, VAT/ASAT, ASAT/GFAT, and VAT/GFAT are shown in parts a), b), c), d), e), and f) respectively. Abbreviations: ASAT (abdominal subcutaneous adipose tissue), GFAT (gluteofemoral adipose tissue), VAT (visceral adipose tissue). SBP (systolic blood pressure), DBP (diastolic blood pressure).
(DOCX)

## Author Contributions

**Conceptualization:** Brian Huang, Michael G. Levin, Victoria M. Gershuni.

**Data curation:** Brian Huang, Michael G. Levin, Victoria M. Gershuni.

**Formal analysis:** Brian Huang, John DePaolo, Renae L. Judy, Gabrielle Shakt, Walter R. Witschey, Michael G. Levin.

**Investigation:** Brian Huang, John DePaolo, Renae L. Judy, Gabrielle Shakt, Walter R. Witschey, Michael G. Levin, Victoria M. Gershuni.

**Methodology:** John DePaolo, Renae L. Judy, Gabrielle Shakt, Walter R. Witschey, Michael G. Levin, Victoria M. Gershuni.

**Project administration:** Victoria M. Gershuni.

**Supervision:** Brian Huang, Michael G. Levin, Victoria M. Gershuni.

**Validation:** Victoria M. Gershuni.

**Visualization:** Victoria M. Gershuni.

**Writing – original draft:** Brian Huang, Walter R. Witschey, Michael G. Levin, Victoria M. Gershuni.

**Writing – review & editing:** Brian Huang, John DePaolo, Renae L. Judy, Gabrielle Shakt, Walter R. Witschey, Michael G. Levin, Victoria M. Gershuni.

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
