## [Decision Letter · Decision Letter 0]

18 Jul 2023

PONE-D-23-16670Relationships between body fat distribution and metabolic syndrome traits and outcomes: a Mendelian Randomization studyPLOS ONE

Dear Dr. Gershuni,

Thank you for submitting your manuscript to PLOS ONE. After careful consideration, we feel that it has merit but does not fully meet PLOS ONE’s publication criteria as it currently stands. Therefore, we invite you to submit a revised version of the manuscript that addresses the points raised during the review process.

Please respond to the minor comments given by one of the reviewers==============================

We look forward to receiving your revised manuscript.

Kind regards,

Fredirick Lazaro mashili, MD, PhD

Academic Editor

PLOS ONE

“J.D. is supported by the American Heart Association (23POST1011251). W.W. is supported by the NIH R01 (HL137984) and NIH P41 (EB029460) grants. M.G.L. is supported by the Institute for Translational Medicine and Therapeutics of the Perelman School of Medicine at the University of Pennsylvania, the NIH/NHLBI National Research Service Award postdoctoral fellowship (T32HL007843), and the Measey Foundation. Research reported in this publication was supported by the National Center for Advancing Translational Sciences of the National Institutes of Health under award number KL2TR001879. The content is solely the responsibility of the authors and does not necessarily represent the official views of the National Institutes of Health.”

“J.D. is supported by the American Heart Association (23POST1011251). W.W. is supported by the NIH R01 (HL137984) and NIH P41 (EB029460) grants. M.G.L. is supported by the Institute for Translational Medicine and Therapeutics of the Perelman School of Medicine at the University of Pennsylvania, the NIH/NHLBI National Research Service Award postdoctoral fellowship (T32HL007843), and the Measey Foundation. Research reported in this publication was supported by the National Center for Advancing Translational Sciences of the National Institutes of Health under award number KL2TR001879. The content is solely the responsibility of the authors and does not necessarily represent the official views of the National Institutes of Health. The funders had no role in study design, data collection and analysis, decision to publish, or preparation of the manuscript.”

Additional Editor Comments:

Please respond to all the comments given by reviewers

Reviewers' comments:

Reviewer's Responses to Questions

**Comments to the Author**

1. Is the manuscript technically sound, and do the data support the conclusions?

Reviewer #1: Yes

Reviewer #2: Yes

2. Has the statistical analysis been performed appropriately and rigorously? 

Reviewer #1: Yes

Reviewer #2: Yes

3. Have the authors made all data underlying the findings in their manuscript fully available?

Reviewer #1: Yes

Reviewer #2: Yes

4. Is the manuscript presented in an intelligible fashion and written in standard English?

Reviewer #1: Yes

Reviewer #2: Yes

5. Review Comments to the Author

Reviewer #1: Relationships between body fat distribution and metabolic syndrome traits and outcomes: a Mendelian Randomization study

The manuscript presents a comprehensive Mendelian Randomization (MR) study exploring the relationships between body fat distribution and traits or outcomes associated with metabolic syndrome. The authors have used genetic variants associated with different body fat distributions, such as visceral adipose tissue (VAT), abdominal subcutaneous adipose tissue (ASAT), and gluteofemoral adipose tissue (GFAT), identified from a Genome-Wide Association Study (GWAS) performed with the United Kingdom BioBank. The methods applied, including two-sample MR and BMI-controlled multivariable MR (MVMR), are robust, offering robust ways to explore the causal relationship between genetic determinants of fat distribution and metabolic syndrome traits. The use of summary statistics from the IEU Open GWAS Project also underscores the broad base of data used to draw conclusions.

The findings suggest a protective cardiometabolic profile associated with increased GFAT, which is maintained in analyses controlling for BMI.

The methodology section of the study provides a detailed description of the data sources, genetic instruments, and statistical approaches used in the research. The application of GWAS in identifying genetic variants associated with different fat distributions, the extraction of GWAS summary statistics for traits and outcomes related to metabolic syndrome, and the implementation of two-sample MR and multivariable MR are thoroughly explained.

Moreover, sensitivity analyses such as weighted median and MR-Egger have been included to strengthen the validity of the findings. This is commendable as these methods can provide more robust estimates under different assumptions about the presence of pleiotropy, which is a limitation in MR studies. The following could be considered to improve the manuscript.

1. One area of the methodology that could have been elaborated is the specifics of the deep learning model used in predicting the ASAT, GFAT, and VAT volumes in the remaining patients. The authors mention a high r² on the hold-out test set but additional information about the type of the model, its architecture, and the performance metrics used would provide a better understanding of its validity.

2. It is important to note that the relationship between increased GFAT and lower levels of LDL-c, Apo-B, total cholesterol, and triglycerides, as well as higher HDL-c and Apo-A1 levels, might not necessarily mean that increasing GFAT will improve these cardiometabolic outcomes. The causality and the mechanisms behind these associations are not addressed by this study and remain to be clarified. The authors should either address this concern by either discussing it or writing it as a limitation.

3. The data show that increased ASAT was significantly associated with increased Type 2 diabetes mellitus (T2DM) risk. However, the study could not determine whether this association is due to the adverse metabolic effects of ASAT per se or because ASAT is simply a marker for overall adiposity or other metabolic dysfunctions. This could be discussed in the discussion section or included as a limitation.

4. One key point is that the results show genetic associations rather than being indicative of changes over time or in response to interventions. So, while GFAT is associated with certain health outcomes, it may not be a modifiable risk factor. The authors should modify their conclusion about causation.

5. The study findings indicate that absolute fat depots and fat depot ratios have differing effects on cardiometabolic health. However, the study does not explore the biological or physiological reasons behind these differences. Will be valuable to discuss the biological or physiological reasons behind these differences based on available evidence/literature.

6. The study's findings are based on genetic data and might not account for environmental factors, such as diet and physical activity, which are known to affect both adiposity and cardiometabolic outcomes. This needs to be discussed or added as a potential limitation.

7. The authors rightly mention that the GWAS utilized consists mainly of individuals aged 40-69 and self-identified as Caucasian. This limits the broad applicability of the results, considering genetic diversity and variance in obesity patterns across different ethnic and age groups. The authors should emphasize this limitation and recommend further studies involving more diverse cohorts.

8. The authors mention that central adiposity measures (VAT and ASAT) did not show robust associations with metabolic syndrome markers in their study, which seems to contradict a significant amount of literature highlighting the deleterious effects of central adiposity. While they point out that their measures of VAT and ASAT might not have captured the full extent of abdominal adiposity, this should be investigated in more detail, and additional explanations should be considered.

9. While the authors summarize the results of other related studies, it would add weight to their arguments if they compared and contrasted their findings more explicitly with the results of these studies.

Reviewer #2: The study is distinctive and novel. Mendelian randomization (MR) approach was employed by the author to investigate the causal relationship between various body fat distributions and the cardio-metabolic syndrome. Publicly available GWAS summary statistics for  different cardio-metabolic syndrome traits and  body fat distribution from IEU and  UK Biobank respectively  were used.

Large gluteal femoral adiposity was found to be cardioprotective and the association is causal, which support findings from observational studies.

6. PLOS authors have the option to publish the peer review history of their article (what does this mean?). If published, this will include your full peer review and any attached files.

Reviewer #1: **Yes: **Fredirick Mashili

Reviewer #2: No

---

## [Author Response · Author response to Decision Letter 0]

11 Sep 2023

We have attached a separate file with our response to reviewers as directed. Thank you for your time.

---

## [Decision Letter · Decision Letter 1]

4 Oct 2023

Relationships between body fat distribution and metabolic syndrome traits and outcomes: a Mendelian Randomization study

PONE-D-23-16670R1

Dear Dr. Gershuni,

We’re pleased to inform you that your manuscript has been judged scientifically suitable for publication and will be formally accepted for publication once it meets all outstanding technical requirements.

Kind regards,

Fredirick Lazaro mashili, MD, PhD

Academic Editor

PLOS ONE

Additional Editor Comments (optional):

All raised concerns have been thoroughly addressed

Reviewers' comments:

Reviewer's Responses to Questions

**Comments to the Author**

1. If the authors have adequately addressed your comments raised in a previous round of review and you feel that this manuscript is now acceptable for publication, you may indicate that here to bypass the “Comments to the Author” section, enter your conflict of interest statement in the “Confidential to Editor” section, and submit your "Accept" recommendation.

Reviewer #1: All comments have been addressed

Reviewer #2: All comments have been addressed

2. Is the manuscript technically sound, and do the data support the conclusions?

Reviewer #1: Yes

Reviewer #2: Yes

3. Has the statistical analysis been performed appropriately and rigorously? 

Reviewer #1: Yes

Reviewer #2: Yes

4. Have the authors made all data underlying the findings in their manuscript fully available?

Reviewer #1: Yes

Reviewer #2: Yes

5. Is the manuscript presented in an intelligible fashion and written in standard English?

Reviewer #1: Yes

Reviewer #2: Yes

6. Review Comments to the Author

Reviewer #1: (No Response)

Reviewer #2: The authors have addressed all the reviewers comments. The paper is due for acceptance and publication.

7. PLOS authors have the option to publish the peer review history of their article (what does this mean?). If published, this will include your full peer review and any attached files.

Reviewer #1: **Yes: **Fredirick Mashili

Reviewer #2: **Yes: **Ikunda Dionis Mushi

---

## [Editor Report · Acceptance letter]

13 Oct 2023

PONE-D-23-16670R1 

Relationships between body fat distribution and metabolic syndrome traits and outcomes: a Mendelian Randomization study. 

Dear Dr. Gershuni:

I'm pleased to inform you that your manuscript has been deemed suitable for publication in PLOS ONE. Congratulations! Your manuscript is now with our production department. 

Kind regards, 

on behalf of

Dr Fredirick Lazaro mashili 

Academic Editor

PLOS ONE